# Predictive Pipelined Decoding:
# A Compute-Latency Trade-off for Exact LLM Decoding

**Seongjun Yang\***
*KRAFTON*
*seongjunyang@krafton.com*

**Gibbeum Lee\***
*KRAFTON*
*pirensisco@krafton.com*

**Jaewoong Cho**
*KRAFTON*
*jwcho@krafton.com*

**Dimitris Papailiopoulos**
*University of Wisconsin-Madison*
*dimitris@papail.io*

**Kangwook Lee**
*University of Wisconsin-Madison*
*KRAFTON*
*kangwook.lee@wisc.edu*

**Reviewed on OpenReview:** *https://openreview.net/forum?id=yUmJ483OBO*

## Abstract

This paper presents Predictive Pipelined Decoding (PPD), an approach that speeds up decoding in Large Language Models (LLMs) while maintaining the exact same output as the original decoding. Unlike conventional strategies, PPD employs additional compute resources to parallelize the initiation of subsequent token decoding during the current token decoding. This method reduces decoding latency and reshapes the understanding of trade-offs in LLM decoding strategies. We have developed a theoretical framework that allows us to analyze the trade-off between computation and latency. Using this framework, we can analytically estimate the potential reduction in latency associated with our proposed method, achieved through the assessment of the match rate, represented as $p_{\text{correct}}$. The results demonstrate that the use of extra computational resources has the potential to accelerate LLM decoding. Additionally, we implement PPD and conduct preliminary experiments to empirically validate its efficacy, addressing potential practical overheads not covered by theoretical analysis.

## 1 Introduction

The recent advances in LLMs, especially transformers (Vaswani et al., 2017), have brought a breakthrough to the domain of natural language processing. The notable generative language models include GPT-3 (Brown et al., 2020), GPT-4 (OpenAI, 2023), PaLM (Chowdhery et al., 2023), LaMDA (Thoppilan et al., 2022), OPT (Zhang et al., 2022), and LLaMA (Touvron et al., 2023a). The power of LLMs is primarily driven by their enormous scale, often involving hundreds of billions of parameters (Hoffmann et al., 2022). However, the considerable size of these models can present challenges in practical applications where immediate responses are crucial (Kasai et al., 2021).

Generative transformers usually produce text sequences using auto-regressive decoding. After passing through all layers of the transformer, each token is generated using the hidden representation from the final layer of the transformer (Kim et al., 2023a). For faster decoding, some studies (Schuster et al., 2022; Tang et al., 2023) have proposed approaches that aim for similar decoding results by only using a sub-network

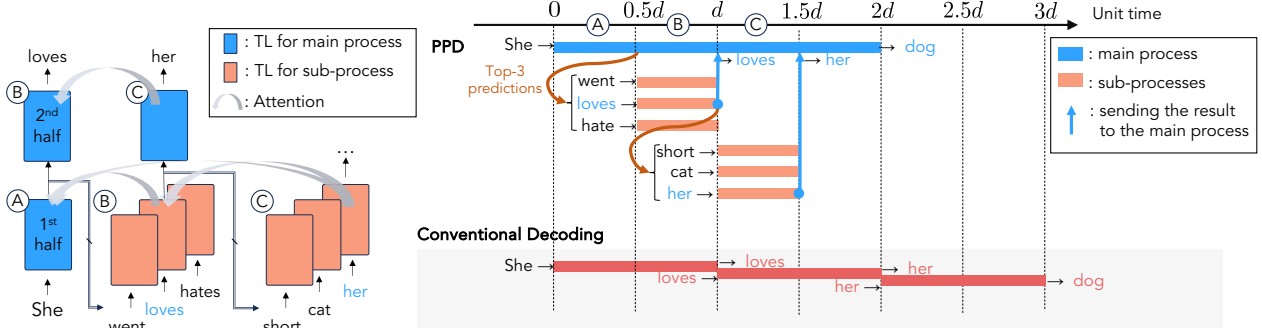

Figure 1: **An overview of the proposed method**. In a scenario where three words are generated from a pre-trained transformer decoder with $d$ layers, "She" is fed as an input for the decoder. PPD forecasts the next token at an intermediate transformer layer, such as the top-3 tokens from the $d/2$-th layer. PPD simultaneously launches three sub-processes, each feeding a predicted token into the model, while the main process continues to forward the intermediate output to the final layer. Once the main process is complete, PPD verifies if any predicted tokens match the main process's final output. If a match is found, this method reduces decoding latency, yielding results equivalent to those of conventional decoding methods. "TL" stands for transformer layers.

of the transformer's layers. However, these approaches do not guarantee the same decoding results when using all layers.

In this paper, we introduce *Predictive Pipelined Decoding (PPD)*, a new approach that lowers latency by utilizing additional compute resources, while keeping the exact same decoding results, as illustrated in Figure 1. Our method is inspired by an approach called *early-exiting*, specifically as described by Schuster et al. (2022). Early-exiting allows the generation process to exit before reaching the final layer, enabling predictions to be made earlier in the process. PPD shares similarities with early exit strategies as it also utilizes intermediate representations to make early predictions on the next token. However, PPD distinguishes itself by continuing the current token decoding without exiting. In other words, the main process remains focused on the current token while other subprocesses early start the generation process with the predicted next token(s).

PPD accelerates decoding by parallelizing processes, each of which begins decoding from the top-$k$ token predictions of the specific transformer layer. Simultaneously, the main process continues to compute the output of the final layer and predicts the next token. By aligning the results with the next token prediction from the final layer, we can maintain the original decoding result.

To assess the potential benefits of our method, we analyze to determine the extent of latency reduction and the associated compute resource costs. Also, we measure the match rate, the probability that the early top-$k$ predictions match the prediction from the final layer, with the commonly utilized dataset in NLP such as SQUAD 1.1 (Rajpurkar et al., 2016), WMT EN-FR (Bojar et al., 2015), and CNN/DM (Hermann et al., 2015). Furthermore, we implement PPD in a multi-GPU setting and show that PPD indeed increases the decoding speed. Note that our primary focus is on providing performance modeling and theoretical analysis. The empirical results from our initial implementation are intended as preliminary evidence to support our theoretical findings.

In summary, our main contributions are the following: (1) a framework, which we call PPD, that boosts the speed of the decoding, (2) a theoretical analysis of latency savings versus computing resource costs, and (3) an implementation to show how effective PPD would be in an actual situation.

## 2 Related Work

Various strategies have been proposed to improve the inference speed of large-scale transformer models. These include employing model pruning techniques (Fan et al., 2019; Gale et al., 2019; Michel et al., 2019; Voita et al., 2019; Sanh et al., 2020; Kurtic et al., 2022; Kwon et al., 2022; Campos & Zhai, 2023); implementing

knowledge distillation methods to downsize the models (Jiao et al., 2020; SANH et al.); and adopting quantization procedures (Zafrir et al., 2019; Shen et al., 2020; Zadeh et al., 2020; Kim et al., 2021; Dettmers et al., 2022; Wu et al., 2022; Yao et al., 2022; Frantar et al., 2023). However, these approaches do not necessarily guarantee the original inference quality since they do not have a mechanism that verifies the validity of the generated token.

Our research is inspired by *early-exiting* approaches (Liu et al., 2021; Schuster et al., 2021; Sun et al., 2022; Xin et al., 2021; Yin et al., 2022; Schuster et al., 2022) that utilize only the initial segments of transformer layers for inference, rather than the entire network. Especially, Schuster et al. (2022) implements an early-exiting approach for decoder-only models in which one can select the layer to exit and check the confidence measure of each token using a threshold function. However, the approach can not be as exact as conventional decoding due to its dependency on a threshold-based confidence measure.

Similarly, with the goal of reducing the inference time of transformers, numerous studies (Kim et al., 2023b; Chen et al., 2023; Leviathan et al., 2023) have utilized two language models which are one smaller and one larger. The smaller model rapidly generates output, while the larger model verifies its validity. Despite the potential speed advantage, this method might not consistently match the exact output of larger models, resulting in discrepancies since the larger model relies on the smaller model's confidence score.

## 3 Predictive Pipelined Decoding

We introduce Predictive Pipelined Decoding (PPD), a low-latency decoding method that leverages multiple compute resources. PPD utilizes an intermediate output of a transformer to predict the next token, which is typically produced by the final layer's output. This allows PPD to start the forward propagation of the next sequence earlier than the conventional decoding. Despite this early start, the original forward propagation continues uninterrupted up to the final layer. This parallel approach accelerates the conventional greedy decoding process while ensuring the same decoding result.

In the following, we elaborate on the process of PPD. This method predicts the next token early at an intermediate transformer layer. PPD employs an intermediate hidden representation $h$, e.g., $\frac{d}{2}$-th layer's output, to estimate the probability $p(x|h)$ of the next token. This is done by applying a language modeling classifier and a softmax activation to the hidden representation. Subsequently, PPD identifies the top-$k$ candidate tokens with regard to $p(x|h)$ and initiates $k$ parallel sub-processes. Each sub-process then inputs the selected token into the transformer. In parallel, the main process continues to forward the intermediate output up to the final layer.

Once the main process completes the forward propagation to the final layer, PPD checks if the decoded next token from the final output matches any of the top-$k$ next token candidates previously predicted from the intermediate output. If a match is found, PPD only continues the forward propagation associated with the matching token, disregarding the results from other processes. In cases where no matches are found, all results from sub-processes are disregarded, and PPD proceeds with the output from the final layer. This approach enables us to achieve decoding results identical to those of the original method while improving latency efficiency. Figure 1 provides an overview of the proposed method. In subsequent rounds, the main process repeatedly employs the output from the intermediate layer of sub-processes. For the sake of clarity, we describe the algorithm under the assumption that $d$ is even and the intermediate layer is set to $d/2$ in Algorithm 1. For more general cases, the algorithm is detailed in Appendix C.

## 4 Theoretical Analysis

### 4.1 Model

For fixed $k$, PPD makes an early next-token prediction at the $\bar{d}$-th intermediate layer out of the total $d$ layers in a transformer. We model that one of the top-$k$ candidates from the early prediction will match the actual top-1 at the final layer with probability $0 < p_{\text{correct}} < 1$. Furthermore, we model that these events, occurring at each token prediction, are independent of one another. We define a sequence of consecutively generated tokens as a *run*. PPD begins generating a *run* and continues until all candidates from the early

---

**Algorithm 1** Predictive Pipelined Decoding (PPD)

---

1: **Input:** maximum number of tokens $\ell$, number of decoder layers $d$, intermediate layer number $d/2$, number of compute units $k+1$, start token $x_0$
2: Launch main process (PID=0) and sub-processes (PID=$1, \ldots, k$)
3: Define $h_{\bar{d}}^{(i)}$ as the hidden representation of the $\bar{d}$-th layer in the process with PID=$i$
4: Initialize:
5:     $t \leftarrow 0$
6:     match $\leftarrow$ False
7: **while** $t < \ell$ and $x_t \neq$ EOS **do**
8:     **for** main process **do**
9:        **if** match $=$ False **then**
10:           Start forwarding from the 1st layer with $x_{\leq t}$ to compute $h_{d/2}^{(0)}$
11:        **end if**
12:        Select the top-$k$ early predicted tokens $\hat{x}_{t+1}^{(1)}, \ldots, \hat{x}_{t+1}^{(k)}$ from $h_{d/2}^{(0)}$
13:        Distribute the early predicted tokens $\hat{x}_{t+1}^{(1)}, \ldots, \hat{x}_{t+1}^{(k)}$ to sub-process $1, \ldots, k$, respectively
14:     **end for**
15:     **for** PID $= 0, 1, ..., k$ **in parallel do**
16:        match $\leftarrow$ False
17:        **if** main process **then**
18:           Start forwarding from the $(d/2+1)$-th layer with $h_{d/2}^{(0)}$
19:           $x_{t+1} \leftarrow$ prediction from the final layer using $h_d^{(0)}$
20:        **else**
21:           Start forwarding from the 1st layer with $(x_{\leq t}, \hat{x}_{t+1}^{(\text{PID})})$ to compute $h_{d/2}^{(\text{PID})}$
22:           **if** $\hat{x}_{t+1}^{(\text{PID})} = x_{t+1}$ **then**
23:              match $\leftarrow$ True
24:              Send $h_{d/2}^{(\text{PID})}$ to main process: $h_{d/2}^{(0)} \leftarrow h_{d/2}^{(\text{PID})}$
25:           **end if**
26:        **end if**
27:     **end for**
28:     $t \leftarrow t+1$
29: **end while**

---

prediction no longer match the final prediction, at which point all sub-processes are disregarded. Counting from the beginning of the generated token, we denote the $i$-th run's length by $X_i$, where $X_i \geq 1$. Note that $X_i \sim \text{Geom}(1 - p_{\text{correct}})$ except for the last run, where Geom denotes the geometric distribution, and $\mathbb{E}[X_i] = 1/(1 - p_{\text{correct}})$. Assume that the length of the text to be generated is $\ell$ tokens, where $\ell \geq 1$. Then, we have $\sum_{i=1}^{N} X_i = \ell$, where $N$ is a random variable that denotes the number of runs required to completely generate $\ell$ tokens. We assume an equal amount of computational time is required for each layer of the transformer, which is mainly due to the majority of layers being composed of the same self-attention and feed-forward network. We refer to this consistent time requirement for one layer as one 'time unit'. Consequently, forward propagation through $d$ layers of the transformer demands $d$ time units.

### 4.2 Latency Analysis

Before delving into our exact analysis, we first present an approximate analysis for $\ell \gg 1$ and $\bar{d} = d/2$ (i.e., making an early prediction at the middle layer).

Let us first find the expression for $N$. Since $\ell \gg 1$, we also have $N \gg 1$. Thus, we have

$$\ell = N \cdot \frac{X_1 + X_2 + \cdots + X_N}{N} \approx N \mathbb{E}[X_1], \tag{1}$$

where the last approximation is derived from the law of large numbers, with the assumption that $X_i$s are i.i.d.

Now, we compute the expected latency to generate $\ell$ tokens with PPD. For a run of length $X$, it takes $d + (X-1)\frac{d}{2} = \frac{d(X+1)}{2}$ time units to generate the run. Please refer to Figure 1. Thus, the total time to generate the $\ell$ tokens is

$$\sum_{i=1}^{N} \frac{d(X_i + 1)}{2} = \frac{d(\sum_{i=1}^{N} X_i + N)}{2} = \frac{d(\ell + N)}{2}. \tag{2}$$

By dividing the total latency by $\ell$, we get the per-token latency:

$$\frac{d(\ell + N)}{2\ell} = \frac{d(1 + N/\ell)}{2} \approx \frac{d(1 + 1/\mathbb{E}[X_1])}{2}$$
$$= d\left(1 - \frac{p_{\text{correct}}}{2}\right). \tag{3}$$

This reveals an intuitive relationship between the per-token latency and the probability of successful early token prediction. If $p_{\text{correct}}$ is close to 1, then the per-token latency becomes $0.5d$, while if $p_{\text{correct}}$ is close to 0, then the average per-token latency remains as $d$.

To run PPD with a fixed choice of $k$, one needs $k + 1$ compute resources. However, at the beginning of each run, only one compute resource is used. Thus, to compute the average compute resources required for running PPD, we need the following calculation. For a run of length $X$, the first $\frac{d}{2}$ time units requires one compute resource, while the remaining $\frac{Xd}{2}$ time units use $k + 1$ compute resources. Therefore, the total compute resources spent for the run of length $X$ is $\frac{(k+1)dX+d}{2}$, and the total compute resources spent for the generation of the $N$ runs is

$$\sum_{i=1}^{N} \frac{(k+1)dX_i + d}{2} = \frac{(k+1)d\ell + dN}{2}. \tag{4}$$

By dividing the total compute resources by the total generation time, we get the average compute resources per time unit:

$$\frac{\frac{(k+1)d\ell+dN}{2}}{\frac{d(\ell+N)}{2}} \approx \frac{(k+1) + 1/\mathbb{E}[X_1]}{1 + 1/\mathbb{E}[X_1]} = \frac{k + 2 - p_{\text{correct}}}{2 - p_{\text{correct}}}. \tag{5}$$

If $p_{\text{correct}}$ is close to 1, then the average compute resources per time unit becomes $k + 1$. Note that this makes sense since when $p_{\text{correct}}$ is 1, one can generate the whole text in one run, and hence all $k+1$ compute units will be used almost all the time. If $p_{\text{correct}}$ is close to 0, then the average compute units per time unit becomes $\frac{k+2}{2}$. This also makes sense as if the early prediction is always wrong, the run length is always 1. For the first half unit time, we use one compute unit. For the second half unit time, we use $k + 1$ compute units. Thus, on average, we use $\frac{k+2}{2}$ compute units throughout the generation.

Recall that the above calculations are the approximate analysis, assuming that $\ell \gg 1$ and $\bar{d} = d/2$. The following theorem provides the exact analysis without the assumption, when $\bar{d} \geq d/2$.

**Theorem 4.1** (Latency-compute trade-off with PPD). *Given $p_{\text{correct}}$, $k$, and for fixed $\ell$, if PPD makes an early prediction at the $\bar{d}$-th intermediate layer among $d$ layers ($\bar{d} \geq d/2$), then the expected latency to generate a sequence of $\ell$ tokens is*

$$d\ell - (d - \bar{d})(\ell - 1)p_{\text{correct}},$$

*and the expected total compute units is*

$$d\ell - (d - \bar{d})(\ell - 1)p_{\text{correct}} + k(d - \bar{d})\ell.$$

*Proof.* For a run of length $X$, the time required to generate the run $T_X$ is given by

$$T_X = d + (X - 1)\bar{d}.$$

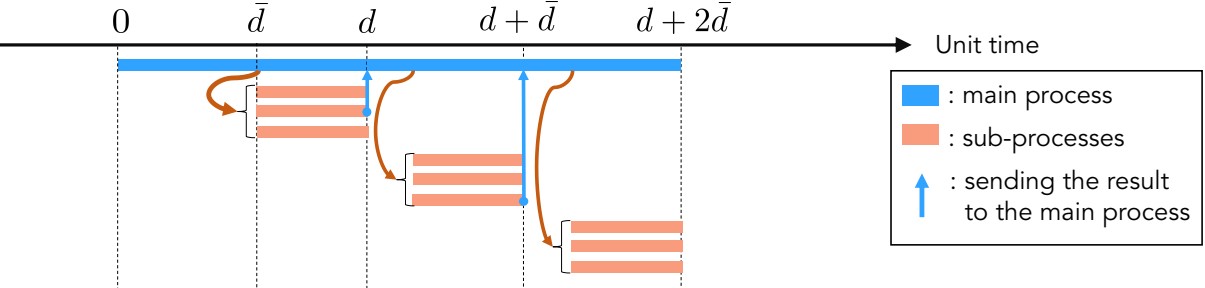

Figure 2: **General flow chart of PPD.** The given figure represents the total latency and usage of compute resources when the run's length $X$ and sub-processes $k$ are both equal to 3 and the match rate $p_{\text{correct}}$ is 1. Also, we assume $\bar{d} \geq d/2$. With the given setting and assumption, the total latency is $d + 2\bar{d}$. Over the entirety of time, four compute resources are engaged for a duration of $3(d - \bar{d})$. The remaining time makes use of just one compute resource.

Therefore, the total latency is

$$\sum_{i=1}^{N} T_{X_i} = \sum_{i=1}^{N} d + (X_i - 1)\bar{d}$$
$$= \bar{d}\sum_{i=1}^{N} X_i + (d - \bar{d})N$$
$$= \bar{d}\ell + (d - \bar{d})N.$$

To compute the expected value of this quantity without assuming $\ell, N \gg 1$, we first need to identify the distribution of $N$. Note that the expectation here is over different instances of sequence generations. Since $N$ is the number of runs, $N-1$ is the number of times early predictions fail. Thus, $N-1 = \text{Bin}(\ell-1, 1-p_{\text{correct}})$. Hence, $N = 1 + \text{Bin}(\ell - 1, 1 - p_{\text{correct}})$. Thus, $\mathbb{E}[N] = 1 + (\ell - 1)(1 - p_{\text{correct}}) = \ell - (\ell - 1)p_{\text{correct}}$. With this, we have

$$\mathbb{E}\left[\bar{d}\ell + (d - \bar{d})N\right] = \bar{d}\ell + (d - \bar{d})\mathbb{E}[N]$$
$$= \bar{d}\ell + (d - \bar{d})\ell - (d - \bar{d})(\ell - 1)p_{\text{correct}}$$
$$= d\ell - (d - \bar{d})(\ell - 1)p_{\text{correct}}.$$

For a run of length $X$, the $(d - \bar{d})X$ time units require $k + 1$ compute resources while the remaining $T_X - (d - \bar{d})X$ time unit requires one compute resource. Therefore, the total compute resources spent for the run of length $X$ are

$$T_X - (d - \bar{d})X + (k + 1)(d - \bar{d})X = d + (X - 1)\bar{d} - (d - \bar{d})X + (k + 1)(d - \bar{d})X$$
$$= \bar{d}X + d - \bar{d} + k(d - \bar{d})X$$
$$= (\bar{d} + k(d - \bar{d}))X + (d - \bar{d}),$$

and the total compute resources spent for the entire text generation is

$$\sum_{i=1}^{N}(\bar{d} + k(d - \bar{d}))X_i + (d - \bar{d}) = (\bar{d} + k(d - \bar{d}))\ell + (d - \bar{d})N.$$

By computing the expected value of it, we have

$$
\begin{aligned}
\mathbb{E}\left[(\bar{d}+k(d-\bar{d}))\ell+(d-\bar{d})N\right] &= (\bar{d}+k(d-\bar{d}))\ell+(d-\bar{d})\mathbb{E}\left[N\right] \\
&= (\bar{d}+k(d-\bar{d}))\ell+(d-\bar{d})(\ell-(\ell-1)p_{\text{correct}}) \\
&= (\bar{d}+(k+1)(d-\bar{d}))\ell-(d-\bar{d})(\ell-1)p_{\text{correct}} \\
&= (d+k(d-\bar{d}))\ell-(d-\bar{d})(\ell-1)p_{\text{correct}} \\
&= d\ell-(d-\bar{d})(\ell-1)p_{\text{correct}}+k(d-\bar{d})\ell.
\end{aligned}
$$

$\square$

Conventional decoding requires $d\ell$ time units to generate $\ell$ tokens. However, in PPD, there is an expectation that a proportion of $(\ell-1)p_{\text{correct}}$ tokens accurately match the predictions made at the intermediate layer. For these instances, parallel pre-computations up to the $(d-\bar{d})$-th layer result in time savings. Consequently, it allows PPD to reduce the expected latency by $(d-\bar{d})(\ell-1)p_{\text{correct}}$ time units.

| | **Prompt** |
|---|---|
| SQUAD | We have provided context information below. <hr> **{context}** <hr> ### Given this information, please answer the question: **{question}** 
 ### Assistant: |
| WMT EN-FR | ### Instruction: Translate English sentence into French. 
 English: Sounds like a typical rugby club to me. 
 French: Ça m'a l'air d'être un club de rugby typique. # 
 English: At an English university, perhaps... 
 French: Dans une université anglaise, peut-être... # 
 English: **{source_sentence}** 
 French: |
| CNN/DM | # Article 
 **{context}** 

 # Summarize the article 
 ### Assistant: |

Table 1: **Prompts used for three NLP tasks on benchmark datasets.**. We use the above formats of prompts to evaluate the match rate $p_{\text{correct}}$. The terms "context", "question", and "source_sentence" represent the corresponding inputs for each task.

To achieve these savings, PPD employs one computational unit dedicated to the main process for $d\ell-(d-\bar{d})(\ell-1)p_{\text{correct}}$ time units. In addition, PPD allocates $k$ computational units for each of the $\ell$ tokens to pre-compute the output of the $(d-\bar{d})$-th layer along with the predictions. Please refer to Figure 2 to help understand the proof.

### 4.3 Simulations

**Experimental Setup** In order to theoretically estimate the potential improvements in decoding latency in real-world NLP tasks, we examine the match rate, denoted by $\hat{p}_{\text{correct}}$. This match rate is empirically estimated across multiple token generation processes by verifying if any of the top-$k$ predicted tokens from the intermediate layer match the top-1 token from the final layer, which is the greedy decoding. Also, we measure the match rate using sampling-based methods such as top-$k$ sampling (Fan et al., 2018) and top-$p$ sampling (Holtzman et al., 2020) in Appendix B

| dataset | $k$ | trained | Layers | | | | |
|---|---|---|---|---|---|---|---|
| | | | 10 | 20 | 30 | 35 | 37 |
| SQUAD | 1 | N | 5.88% | 38.90% | 62.90% | 79.77% | 88.01% |
| | | Y | 15.45% | 52.81% | 72.34% | 87.68% | 91.67% |
| | 3 | N | 9.25% | 54.04% | 77.92% | 92.64% | 97.67% |
| | | Y | 23.48% | 68.37% | 87.49% | 97.33% | 98.91% |
| | 5 | N | 11.04% | 60.15% | 83.84% | 95.85% | 99.08% |
| | | Y | 27.90% | 74.15% | 92.29% | 98.81% | 99.62% |
| WMT | 1 | N | 2.40% | 21.63% | 39.69% | 68.64% | 78.15% |
| | | Y | 11.06% | 29.17% | 48.20% | 74.84% | 82.69% |
| | 3 | N | 4.38% | 31.69% | 61.71% | 85.03% | 93.53% |
| | | Y | 14.83% | 41.14% | 68.50% | 89.84% | 95.48% |
| | 5 | N | 5.57% | 37.13% | 68.84% | 89.54% | 96.41% |
| | | Y | 16.82% | 47.84% | 75.46% | 93.36% | 97.67% |
| CNN/DM | 1 | N | 7.23% | 32.08% | 53.07% | 68.90% | 78.82% |
| | | Y | 19.02% | 43.65% | 61.45% | 78.46% | 84.42% |
| | 3 | N | 12.84% | 46.36% | 68.14% | 85.07% | 93.81% |
| | | Y | 27.57% | 60.60% | 78.55% | 93.07% | 96.62% |
| | 5 | N | 15.21% | 52.51% | 74.22% | 90.04% | 96.88% |
| | | Y | 31.33% | 67.33% | 84.83% | 96.06% | 98.40% |

Table 2: **The result of $\hat{p}_{\mathbf{correct}}$ from Vicuna-13B**. The match rate, $\hat{p}_{\text{correct}}$, represents the probability where one of the top-$k$ predictions from the intermediate layer matches the top-1 prediction from the final layer. In the "trained" column, "N" signifies that the language modeling classifier, which is trained specifically for the final layer, tests across all layers. Conversely, "Y" represents the classifier individually trained for each layer.

We test the NLP tasks on three benchmark datasets: SQUAD 1.1 (Rajpurkar et al., 2016), WMT EN-FR (Bojar et al., 2015), and CNN/DM (Hermann et al., 2015).

To be specific, SQUAD 1.1 Rajpurkar et al. (2016) is a Question Answering dataset that has 10,570 test pairs. WMT15 FR-EN Bojar et al. (2015) is a machine translation dataset that includes 1,500 test pairs of English to French translations. CNN/DM Hermann et al. (2015) is a dataset for text summarization which has 11,487 test pairs. For these datasets, we set the token length for text generation at 16 for SQUAD 1.1, and 128 for both the WMT EN-FR and CNN/DM. We use their respective test datasets for evaluations. The model for the test is Vicuna-13B (Chiang et al., 2023), a transformer with a total of 40 layers.

We evaluate $\hat{p}_{\text{correct}}$ from Vicuna-13B (Chiang et al., 2023) trained by fine-tuning LLaMA (Touvron et al., 2023a) on user-shared dialogues collected from the website ShareGPT[1]. To the best of our knowledge, the model proposed by Chiang et al. (2023) demonstrates one of the highest performances among the 13B-sized open-source models currently available. We conduct text generation using the prompts provided in Table 1 and the test datasets. For each token generated, we compare the early predictions from the intermediate layer to the final output. For example, we base the match rate evaluation on $10,570 \times 16$ comparisons for the SQUAD 1.1 dataset. All experiments are conducted using the Huggingface Transformers library (Wolf et al., 2020). We specifically probe the early prediction at the 15th, 20th, 30th, 35th, and 37th layers to derive the match rate. please refer to Table 2.

Furthermore, our analysis includes two different utilization of the language modeling classifier for estimating the distribution of tokens over vocabulary. The first employs the common classifier across all layers, trained specifically for the final layer, and the second uses the classifier trained for a given intermediate layer. To train the language modeling classifier at an intermediate layer in Vicuna-13B, we employ the ShareGPT

---

[1]https://sharegpt.com/

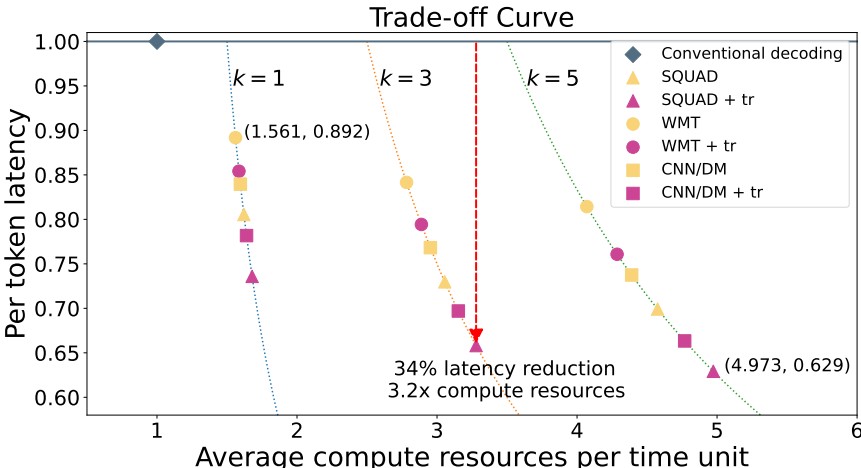

Figure 3: **Theoretical trade-off curve of average compute resources per time unit and per token latency.** The curve graph is derived from Equation 3 and 5. For example, with $k = 3$ and while performing the SQUAD task with a trained classifier, latency can be reduced by 34% at the expense of using 3.2 times more computational resources. This is demonstrated using Vicuna-13B, a model with 40 layers, where the intermediate layer is set to $d/2$. The notation "tr" indicates that the language modeling classifier has been individually trained for each transformer decoder layer.

dataset. We freeze all model parameters excluding those of the language modeling classifier. The training process is performed on 8 A100 GPUs, and the hyperparameters can be found in the Table 4 in Appendix A.

**Result** Table 2 shows the results of $\hat{p}_{\text{correct}}$ from Vicuna-13B. In the table, "N" denotes evaluations using the original pre-trained language modeling classifier, while "Y" indicates evaluations with the fine-tuned language modeling classifier for an intermediate layer. The results show that an increase in either $k$ or the prediction layer number enhances the accuracy of early prediction. Furthermore, employing a language modeling classifier trained for an intermediate layer leads to improvement in accuracy.

We provide the analysis of match rates with respect to token positions in Tables 6 to 8 in Appendix D. Consistent trends in token prediction are observed across various token positions when considering a specific layer, $k$, and language modeling classifier.

Figure 3 illustrates the theoretical trade-off between latency and computational resources, derived from Equation 3 and 5. The probability $p_{\text{correct}}$ applied in the curve is based on $\hat{p}_{\text{correct}}$ values at the 20-th intermediate layer, as presented in Table 2. With respect to the conventional greedy decoding, the curve shows that normalized latency per token ranges from 0.629 (SQUAD+"tr", $k$=5) to 0.892 (WMT, $k$=1). This indicates potential latency improvements of between 10.8% and 37.1%, while preserving the output quality comparable to the original decoding. However, it is important to note that these speed gains come at the cost of additional computational resources, which can increase by a factor of 1.561 to nearly 4.973 times the original consumption. For further insights into the trade-off between average computational resources per token and latency, we refer the reader to Figure 4. We illustrate the trade-off between latency and compute resources per token for $\bar{d} = d/2$. The "compute resources per token" is calculated by multiplication of Equation 3 and 5:

$$\text{Average compute resources per token} \approx \frac{2 + k - p_{\text{correct}}}{2}. \tag{6}$$

## 5 Implementation

We implement PPD and conduct some preliminary experiments to provide empirical evidence supporting the efficacy of the method since the theoretical analysis does not fully address the potential overheads that may invalidate the latency gains in practical applications. To measure the latency, we employ the LLaMA-2

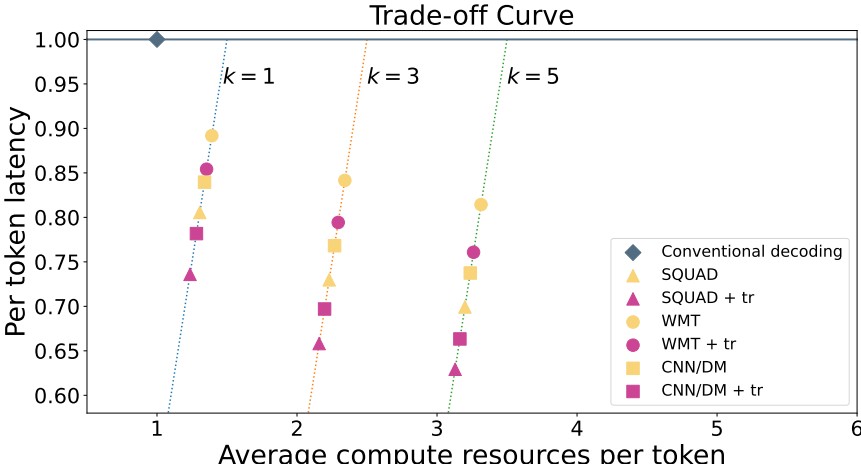

Figure 4: **Theoretical trade-off curve of compute resources per token and latency.** In this figure, we change the x-axis of Figure 3, average compute resources per time unit, to average compute resources per token. The model and experimental setting are the same as those used in Figure 3.

| Method | $k$ | CNN/DM | | | SQUAD 1.1 | | |
| --- | --- | --- | --- | --- | --- | --- | --- |
| | | $\hat{p}_{\text{correct}}$ | Latency ↓ | Throughput ↑ | $\hat{p}_{\text{correct}}$ | Latency ↓ | Throughput ↑ |
| greedy | | - | 18.171 | 7.044 | - | 14.994 | 8.537 |
| greedy (w/ *PPD*) | 1 | 25.72 % | 17.019 | 7.521 | 30.97 % | 13.711 | 9.336 |
| greedy (w/ *PPD*) | 3 | 41.99 % | 16.685 | 7.671 | 47.24 % | 13.712 | 9.335 |

Table 3: **The implementation result.** The letter $k$ represents the number of subprocesses or the amount of additional GPU resources. Latency denotes the time taken to generate 128 tokens, measured in seconds. Throughput signifies the number of tokens generated per second.

(Touvron et al., 2023b) 13B model and conduct tests on several examples from Summarization (CNN/DM) and Question Answering (SQUAD 1.1) tasks. We compare the speed of greedy decoding without PPD and that of greedy decoding with PPD. Additionally, we calculate the average latency and throughput when the language model generates 128 tokens.

From the results of Table 3, we observe that PPD potentially operates at a faster speed compared to the greedy decoding without PPD. However, there is no gain in latency when we compare the results of $k$=1 to that of $k$=3 in the SQUAD 1.1 task due to the communication cost between processes. Building on these findings, our future research will focus on mitigating the communication costs to further enhance the performance of PPD. The code for our implementation is available in the supplementary material.

## 6 Limitations

While our method has the potential for latency improvements, this comes at the cost of increased computational requirements. To reduce the computation costs, future research should focus on better utilization of GPU resources. It is also crucial to consider other factors that impact latency, such as GPU synchronization, data transfer overhead, and communication and memory management overhead, as highlighted in Kim et al. (2023a). The scope of our current work specifically targets greedy decoding, yet it is worth acknowledging that other decoding algorithms (Holtzman et al., 2020; Radford et al., 2019; Wang et al., 2023) have demonstrated superior performance. Therefore, we measure the $\hat{p}_{\text{correct}}$ of other decoding algorithms to evaluate the potential speed enhancements with PPD. The corresponding results are available in Appendix B. Thus, future endeavors intend to extend our methodology to other decoding methods.

## 7 Conclusion

We introduced PPD, a method aimed at reducing the decoding latency while maintaining the original decoding result of LLM. Based on our theoretical analysis and empirical measurements, we identified the potential of PPD to reduce latency. Furthermore, we demonstrated that training the language modeling classifier for an intermediate transformer layer can effectively enhance early prediction accuracy, potentially leading to further reductions in latency.

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

# A  Experiment Setting

| Hyperparameter | Value |
|---|---|
| Number of Epochs | 3 |
| Learning Rate | 0.00002 |
| Batch Size | 128 |
| Optimizer | AdamW |
| Loss Function | Cross-Entropy |
| Max Sequence Length | 2048 |
| Warmup ratio | 0.04 |
| Weight Decay | 0.0 |

Table 4: Hyperparameters used for training a language modeling classifier for an intermediate layer.

# B  The match rate, $\hat{p}_{\text{correct}}$, of sampling based methods

| Method | CNN/DM | SQUAD 1.1 |
|---|---|---|
| Greedy decoding | 40.84% | 50.77% |
| Top-$k$ sampling | 40.45% | 43.78% |
| Top-$p$ sampling | 40.16% | 43.06% |

Table 5: The result of $\hat{p}_{\text{correct}}$ from sampling based methods.

In this section, we clarify that sampling-based decoding methods, such as top-$k$ sampling (Fan et al., 2018) and top-$p$ sampling (Holtzman et al., 2020), can also utilize PPD. The only difference from the application to greedy decoding is that a final prediction is selected through top-$k$ or top-$p$ sampling, rather than using the greedy decoding. We compare the $\hat{p}_{\text{correct}}$ of three algorithms - 1) greedy, 2) top-$k$, and 3) top-$p$ - by checking if any of the top-$\hat{k}$ predicted tokens from the intermediate layer correspond with the decoding result token from the final layer. Here, we employ the language modeling classifier which is trained specifically for the final layer.

In our experimental setup, we randomly sample 100 examples from the CNN/DM and SQUAD 1.1 datasets using the LLaMA-2 (Touvron et al., 2023b) 13B model. We utilize three processes ($\hat{k} = 3$) to compare the performance of the greedy, top-$k$, and top-$p$ decoding algorithms. For top-$k$ sampling, we set $k$ to 3, and for top-$p$ sampling, we set $p$ to 0.95.

The $\hat{p}_{\text{correct}}$ comparison between greedy and other methods (top-$k$, top-$p$) shows insignificant differences in CNN/DM. Despite minor variations in SQAUD 1.1, non-greedy methods maintain a high match rate. These results suggest that applying PPD to the sampling-based decoding method can potentially enhance decoding speed.

## C  PPD Algorithm in General Cases

---

**Algorithm 2** Predictive Pipelined Decoding (PPD)

---

1: **Input:** maximum number of tokens $\ell$, number of decoder layers $d$, intermediate layer number $\bar{d}(\geq d/2)$, number of compute units $k+1$, start token $x_0$
2: Launch main process (PID=0) and sub-processes (PID=1, . . . , k)
3: Define $h_{\bar{d}}^{(i)}$ as the hidden representation of the $\bar{d}$-th layer in the process with PID=$i$
4: Initialize:
5:      $t \leftarrow 0$
6:      match $\leftarrow$ False
7: **while** $t < \ell$ **and** $x_t \neq$ EOS **do**
8:      **for** main process **do**
9:          **if** match = False **then**
10:              Start forwarding from the 1st layer with $x_{\leq t}$ to compute $h_{\bar{d}}^{(0)}$
11:          **else**
12:              Start forwarding from the $(d - \bar{d} + 1)$-th layer with $(x_{\leq t}, h_{d-\bar{d}}^{(0)})$ to compute $h_{\bar{d}}^{(0)}$
13:          **end if**
14:          Select the top-$k$ early predicted tokens $\hat{x}_{t+1}^{(1)}, \ldots, \hat{x}_{t+1}^{(k)}$ from $h_{\bar{d}}^{(0)}$
15:          Distribute the early predicted tokens $\hat{x}_{t+1}^{(1)}, \ldots, \hat{x}_{t+1}^{(k)}$ to sub-process 1, . . . , k, respectively
16:      **end for**
17:      **for** PID = 0, 1, ..., k **in parallel do**
18:          match $\leftarrow$ False
19:          **if** main process **then**
20:              Start forwarding from the $(\bar{d} + 1)$-th layer with $h_{\bar{d}}^{(0)}$
21:              $x_{t+1} \leftarrow$ prediction from the final layer using $h_d^{(0)}$
22:          **else**
23:              Start forwarding from the 1st layer with $(x_{\leq t}, \hat{x}_{t+1}^{(\text{PID})})$ to compute $h_{d-\bar{d}}^{(\text{PID})}$
24:              **if** $\hat{x}_{t+1}^{(\text{PID})} = x_{t+1}$ **then**
25:                  match $\leftarrow$ True
26:                  Send $h_{d-\bar{d}}^{(\text{PID})}$ to main process: $h_{d-\bar{d}}^{(0)} \leftarrow h_{d-\bar{d}}^{(\text{PID})}$
27:              **end if**
28:          **end if**
29:      **end for**
30:      $t \leftarrow t + 1$
31: **end while**

---

# D   Additional Tables

| $\bar{d}$ | $k$ | Trained | Token position 1−2 | 3−4 | 5−6 | 7−8 | 9−10 | 11−12 | 13−14 | 15−16 | Total |
|---|---|---|---|---|---|---|---|---|---|---|---|
| 10 layers | 1 | N | 0.80% | 7.29% | 6.02% | 5.66% | 5.93% | 6.62% | 7.02% | 7.66% | 5.88% |
| | | Y | 3.06% | 16.79% | 16.76% | 15.45% | 16.58% | 17.72% | 17.98% | 19.28% | 15.45% |
| | 3 | N | 1.43% | 11.74% | 9.31% | 9.23% | 9.70% | 10.53% | 10.61% | 11.40% | 9.25% |
| | | Y | 13.52% | 25.17% | 24.14% | 23.20% | 24.40% | 25.35% | 25.38% | 26.69% | 23.48% |
| | 5 | N | 1.91% | 14.34% | 11.11% | 11.11% | 11.70% | 12.38% | 12.52% | 13.25% | 11.04% |
| | | Y | 23.42% | 29.57% | 27.36% | 27.04% | 27.87% | 28.92% | 28.85% | 30.14% | 27.90% |
| 20 layers | 1 | N | 19.56% | 43.20% | 44.55% | 43.36% | 41.81% | 40.36% | 39.31% | 39.04% | 38.90% |
| | | Y | 37.54% | 56.01% | 56.53% | 55.93% | 54.96% | 53.92% | 53.69% | 53.93% | 52.81% |
| | 3 | N | 30.15% | 60.79% | 60.86% | 59.67% | 57.75% | 55.23% | 54.35% | 53.52% | 54.04% |
| | | Y | 51.10% | 72.95% | 72.39% | 71.82% | 71.24% | 69.80% | 68.60% | 69.06% | 68.37% |
| | 5 | N | 36.47% | 67.80% | 66.63% | 65.79% | 63.89% | 61.24% | 59.86% | 59.51% | 60.15% |
| | | Y | 57.37% | 79.05% | 78.03% | 77.62% | 76.92% | 75.63% | 74.05% | 74.53% | 74.15% |
| 30 layers | 1 | N | 55.05% | 70.19% | 66.48% | 66.25% | 64.72% | 61.70% | 60.42% | 58.39% | 62.90% |
| | | Y | 65.36% | 78.73% | 76.79% | 74.30% | 73.76% | 71.01% | 69.65% | 69.09% | 72.34% |
| | 3 | N | 71.66% | 85.21% | 81.68% | 80.46% | 79.86% | 76.47% | 74.84% | 73.15% | 77.92% |
| | | Y | 84.03% | 92.37% | 91.04% | 88.73% | 88.43% | 86.06% | 85.05% | 84.20% | 87.49% |
| | 5 | N | 78.84% | 90.15% | 87.94% | 86.12% | 85.43% | 82.22% | 80.77% | 79.30% | 83.84% |
| | | Y | 91.12% | 95.70% | 94.90% | 93.41% | 92.62% | 90.85% | 90.27% | 89.43% | 92.29% |
| 35 layers | 1 | N | 73.73% | 85.51% | 82.93% | 81.13% | 80.75% | 78.99% | 78.10% | 76.98% | 79.77% |
| | | Y | 82.74% | 91.32% | 90.70% | 88.20% | 88.25% | 87.39% | 86.96% | 85.89% | 87.68% |
| | 3 | N | 90.03% | 96.02% | 94.68% | 93.48% | 93.48% | 92.01% | 91.32% | 90.10% | 92.64% |
| | | Y | 96.26% | 98.58% | 98.32% | 97.88% | 97.65% | 97.11% | 96.70% | 96.11% | 97.33% |
| | 5 | N | 94.64% | 98.12% | 97.40% | 96.54% | 96.20% | 95.27% | 94.64% | 94.01% | 95.85% |
| | | Y | 98.55% | 99.44% | 99.38% | 99.19% | 98.87% | 98.71% | 98.37% | 97.98% | 98.81% |
| 37 layers | 1 | N | 82.12% | 91.66% | 90.79% | 88.70% | 88.90% | 87.89% | 87.58% | 86.44% | 88.01% |
| | | Y | 87.70% | 94.28% | 93.73% | 91.95% | 92.28% | 91.66% | 91.29% | 90.50% | 91.67% |
| | 3 | N | 96.84% | 98.86% | 98.57% | 98.09% | 98.02% | 97.55% | 97.04% | 96.37% | 97.67% |
| | | Y | 98.56% | 99.47% | 99.35% | 99.19% | 99.12% | 98.91% | 98.58% | 98.07% | 98.91% |
| | 5 | N | 99.04% | 99.62% | 99.53% | 99.30% | 99.17% | 99.03% | 98.71% | 98.28% | 99.08% |
| | | Y | 99.56% | 99.81% | 99.84% | 99.77% | 99.70% | 99.62% | 99.44% | 99.21% | 99.62% |

Table 6: The token prediction results with respect to token positions in SQUAD.

| $\bar{d}$ | $k$ | Trained | Token position | | | | | | | | Total |
|---|---|---|---|---|---|---|---|---|---|---|---|
| | | | $1-16$ | $17-32$ | $33-48$ | $49-64$ | $65-80$ | $81-96$ | $97-112$ | $113-128$ | |
| 10 layers | 1 | N | 1.45% | 2.01% | 2.35% | 2.12% | 2.35% | 2.66% | 2.93% | 3.30% | 2.40% |
| | | Y | 3.60% | 8.08% | 9.60% | 10.80% | 12.02% | 13.42% | 14.92% | 16.08% | 11.06% |
| | 3 | N | 2.65% | 3.74% | 4.33% | 4.35% | 4.40% | 4.74% | 5.20% | 5.60% | 4.38% |
| | | Y | 5.53% | 11.08% | 13.33% | 14.33% | 15.86% | 17.89% | 19.75% | 20.90% | 14.83% |
| | 5 | N | 3.54% | 4.82% | 5.50% | 5.59% | 5.58% | 6.03% | 6.51% | 7.00% | 5.57% |
| | | Y | 6.73% | 13.00% | 15.05% | 15.92% | 17.84% | 20.24% | 22.38% | 23.43% | 16.82% |
| 20 layers | 1 | N | 11.55% | 16.65% | 17.45% | 20.37% | 22.60% | 25.95% | 28.40% | 30.08% | 21.63% |
| | | Y | 13.60% | 21.05% | 24.03% | 28.37% | 31.07% | 35.65% | 38.65% | 40.95% | 29.17% |
| | 3 | N | 17.87% | 24.17% | 25.78% | 29.95% | 33.10% | 38.50% | 41.23% | 42.89% | 31.69% |
| | | Y | 22.08% | 30.01% | 33.97% | 39.87% | 43.90% | 50.11% | 53.63% | 55.55% | 41.14% |
| | 5 | N | 22.33% | 28.84% | 30.29% | 34.92% | 39.20% | 44.41% | 47.65% | 49.44% | 37.13% |
| | | Y | 27.49% | 36.51% | 40.65% | 46.85% | 51.28% | 57.24% | 60.45% | 62.28% | 47.84% |
| 30 layers | 1 | N | 28.97% | 32.49% | 33.22% | 38.12% | 40.79% | 45.30% | 47.90% | 50.70% | 39.69% |
| | | Y | 32.79% | 39.03% | 40.70% | 46.02% | 50.68% | 55.72% | 58.84% | 61.83% | 48.20% |
| | 3 | N | 48.05% | 53.57% | 55.64% | 60.53% | 64.22% | 67.84% | 70.70% | 73.16% | 61.71% |
| | | Y | 52.70% | 59.63% | 62.35% | 66.78% | 72.00% | 75.52% | 78.53% | 80.46% | 68.50% |
| | 5 | N | 56.83% | 61.02% | 62.43% | 66.80% | 71.14% | 75.01% | 77.78% | 79.71% | 68.84% |
| | | Y | 61.77% | 67.57% | 69.71% | 73.43% | 78.43% | 81.90% | 84.68% | 86.21% | 75.46% |
| 35 layers | 1 | N | 59.78% | 60.45% | 62.65% | 66.26% | 71.51% | 73.71% | 76.38% | 78.43% | 68.64% |
| | | Y | 64.29% | 66.23% | 69.20% | 73.04% | 77.93% | 80.60% | 82.96% | 84.49% | 74.84% |
| | 3 | N | 79.89% | 79.73% | 80.28% | 82.90% | 86.58% | 89.11% | 90.45% | 91.33% | 85.03% |
| | | Y | 83.93% | 86.30% | 86.48% | 88.28% | 91.11% | 93.23% | 94.40% | 94.97% | 89.84% |
| | 5 | N | 85.90% | 86.35% | 85.90% | 87.53% | 90.48% | 92.51% | 93.58% | 94.05% | 89.54% |
| | | Y | 89.22% | 91.40% | 90.74% | 92.10% | 94.12% | 95.72% | 96.65% | 96.90% | 93.36% |
| 37 layers | 1 | N | 73.90% | 71.87% | 72.45% | 75.47% | 80.09% | 81.95% | 84.05% | 85.42% | 78.15% |
| | | Y | 77.30% | 76.32% | 77.35% | 80.82% | 84.78% | 86.85% | 88.49% | 89.58% | 82.69% |
| | 3 | N | 91.86% | 91.50% | 91.50% | 92.25% | 93.83% | 95.14% | 95.84% | 96.34% | 93.53% |
| | | Y | 93.66% | 94.02% | 93.89% | 94.69% | 95.87% | 96.79% | 97.31% | 97.64% | 95.48% |
| | 5 | N | 95.60% | 95.69% | 95.01% | 95.56% | 96.51% | 97.23% | 97.76% | 97.90% | 96.41% |
| | | Y | 96.69% | 97.14% | 96.75% | 97.15% | 97.81% | 98.29% | 98.66% | 98.88% | 97.67% |

Table 7: The token prediction results with respect to token positions in WMT EN-FR.

| $\bar{d}$ | $k$ | Trained | Token position | | | | | | | | |
|---|---|---|---|---|---|---|---|---|---|---|---|
| | | | $1-16$ | $17-32$ | $33-48$ | $49-64$ | $65-80$ | $81-96$ | $97-112$ | $113-128$ | Total |
| 10 layers | 1 | N | 0.85% | 7.42% | 7.43% | 7.99% | 8.26% | 8.64% | 8.71% | 8.54% | 7.23% |
| | | Y | 15.32% | 17.20% | 17.80% | 18.56% | 19.27% | 20.29% | 21.43% | 22.27% | 19.02% |
| | 3 | N | 12.75% | 11.97% | 12.01% | 12.94% | 13.25% | 13.50% | 13.33% | 13.01% | 12.84% |
| | | Y | 21.92% | 25.55% | 26.63% | 27.63% | 28.43% | 29.43% | 30.18% | 30.78% | 27.57% |
| | 5 | N | 14.82% | 14.31% | 14.44% | 15.46% | 15.73% | 16.05% | 15.68% | 15.21% | 15.21% |
| | | Y | 24.98% | 29.34% | 30.45% | 31.61% | 32.41% | 33.36% | 33.99% | 34.53% | 31.33% |
| 20 layers | 1 | N | 33.34% | 33.67% | 31.85% | 31.87% | 31.46% | 31.64% | 31.50% | 31.29% | 32.08% |
| | | Y | 42.24% | 44.45% | 43.03% | 43.09% | 43.16% | 43.75% | 44.50% | 45.01% | 43.65% |
| | 3 | N | 47.68% | 48.47% | 46.01% | 46.23% | 45.82% | 45.96% | 45.64% | 45.03% | 46.36% |
| | | Y | 60.46% | 61.38% | 59.67% | 59.98% | 60.19% | 60.76% | 61.13% | 61.25% | 60.60% |
| | 5 | N | 54.27% | 54.68% | 52.17% | 52.40% | 52.01% | 52.06% | 51.71% | 50.80% | 52.51% |
| | | Y | 67.83% | 68.02% | 66.35% | 66.72% | 66.88% | 67.52% | 67.71% | 67.63% | 67.33% |
| 30 layers | 1 | N | 57.17% | 54.63% | 52.49% | 52.79% | 52.35% | 52.10% | 51.86% | 51.15% | 53.07% |
| | | Y | 63.01% | 62.35% | 60.84% | 61.26% | 61.17% | 61.15% | 61.14% | 60.68% | 61.45% |
| | 3 | N | 70.92% | 69.59% | 67.80% | 68.27% | 68.06% | 67.64% | 67.01% | 65.81% | 68.14% |
| | | Y | 78.99% | 79.32% | 78.63% | 78.98% | 78.85% | 78.58% | 77.97% | 77.05% | 78.55% |
| | 5 | N | 76.21% | 75.51% | 74.03% | 74.51% | 74.39% | 74.05% | 73.18% | 71.88% | 74.22% |
| | | Y | 84.95% | 85.47% | 85.31% | 85.40% | 85.32% | 84.94% | 84.10% | 83.16% | 84.83% |
| 35 layers | 1 | N | 70.56% | 70.11% | 68.56% | 68.49% | 68.45% | 68.41% | 68.40% | 68.20% | 68.90% |
| | | Y | 78.70% | 79.48% | 78.51% | 78.44% | 78.23% | 78.27% | 78.14% | 77.92% | 78.46% |
| | 3 | N | 85.92% | 85.89% | 85.16% | 85.10% | 85.00% | 84.95% | 84.61% | 83.95% | 85.07% |
| | | Y | 93.22% | 93.69% | 93.52% | 93.44% | 93.28% | 93.02% | 92.60% | 91.80% | 93.07% |
| | 5 | N | 90.46% | 90.66% | 90.32% | 90.27% | 90.14% | 89.99% | 89.67% | 88.83% | 90.04% |
| | | Y | 96.33% | 96.56% | 96.49% | 96.41% | 96.24% | 95.99% | 95.59% | 94.82% | 96.06% |
| 37 layers | 1 | N | 79.38% | 79.84% | 78.79% | 78.44% | 78.36% | 78.41% | 78.57% | 78.78% | 78.82% |
| | | Y | 84.51% | 85.32% | 84.51% | 84.30% | 84.14% | 84.12% | 84.17% | 84.27% | 84.42% |
| | 3 | N | 93.72% | 94.31% | 94.07% | 93.97% | 93.78% | 93.79% | 93.57% | 93.28% | 93.81% |
| | | Y | 96.71% | 97.02% | 96.94% | 96.85% | 96.74% | 96.52% | 96.26% | 95.91% | 96.62% |
| | 5 | N | 96.96% | 97.20% | 97.15% | 97.13% | 97.04% | 96.88% | 96.58% | 96.12% | 96.88% |
| | | Y | 98.64% | 98.66% | 98.60% | 98.59% | 98.50% | 98.33% | 98.11% | 97.77% | 98.40% |

Table 8: The token prediction results with respect to token positions in CNN/DM.

