# OpenReview forum: "Predictive Pipelined Decoding: A Compute-Latency Trade-off for Exact LLM Decoding"
_TMLR — Accepted by TMLR_

### Review · Reviewer_WrQN · 2023-11-15

**Summary Of Contributions:**

This paper presents a speculative decoding scheme for LLMs to speed up inference. They use an intermediate output to predict token X, then speculatively begin to decode token X+1 and only commit/use its result if token X was verified by the final LLM output. This results in more computations generally but LLMs are memory bound, meaning that this _could_ result in lower overall latency,

**Audience:**

Yes

**Claims And Evidence:**

No

**Requested Changes:**

- some comparison to prior work (doesn't have to be everything, but a comparison would help to situate this work within the literature)
- Actual implementation (at least PoC) and measurement of actual speedup
- commentary or experiments with non-greedy decoding

**Strengths And Weaknesses:**

Strengths: The idea is interesting and timely.

Weaknesses:
- only based on greedy decoding.
- no comparisons to other speculative decoding methods [1-5].
- no implementation of this so the speedup is only theoreticalnand many assumptions are made about how fast this could actually run with parallel/pipelined decoding occuring in parallel. My guess is that the actual speedup will be far less than what's predicted here and there may even be an overall slowdown in many cases.

[1] Speculative Decoding with big-little decoder
[2] Fast Inference from transformers via speculative decoding
[3] Depth-adaptive Transformers
[4] Content Adaptive Language Modeling
[5] LLMCad: Fast and Scalable On-device Large Language Model Inference

---

> ### Author Response · Authors · 2024-01-17
> **Author's Response to Reviewer WrQN**
>
> We appreciate the constructive critiques. We tried to make our best effort to clarify a few points in this rebuttal.
>
> **Response to "This paper presents a speculative decoding scheme for LLMs to speed up inference. No comparisons to other speculative decoding methods [1-5]."**
>
> We would like to clarify that speculative decoding and PPD should be considered as distinct strategies. This assertion is based on the fact that PPD can be incorporated into the draft model, or small model, of speculative decoding, particularly when there is a need to enhance the speed of speculative decoding further.
>
> The primary distinctions between these strategies lie in the number of models required and the reliability of the output. PPD needs only a single model, whereas speculative decoding requires two distinct models. Moreover, PPD guarantees an output identical to the original decoding, while speculative decoding does not assure a perfect alignment between the outputs of the draft and original models.
>
> **Response to "The paper did not consider non-greedy decoding"**
>
> We would like to emphasize that PPD can also be utilized with other decoding methods, such as top-*k* and top-*p* sampling. The only difference is that a final prediction is selected through top-*k* or top-*p* sampling, rather than using the greedy decoding. We have compared the match rates of three algorithms - 1) greedy, 2) top-*k*, and 3) top-*p* - by checking if any of the top-$\bar{k}$ predicted tokens from the intermediate layer correspond with the decoding result token from the final layer. Here, we employ the language modeling classifier which is trained specifically for the final layer.
>
> Experimental setup:
> - Tasks: QA (Squad 1.1), Summarization (CNN/DM)
> - Model: Llama-2-13b-chat
> - Randomly sampled 100 examples
> - Number of processes ($\bar{k}$): 3
>
> The match rate (%):
>
> |      Decoding  Method   | Summarization | Question Answering |
> | --------       | --------      | --------           |
> | Greedy decoding         | 40.84 %       | 50.77 %            |
> | Top-*k* sampling (*k*=3)    | 40.45 %       | 43.78 %            |
> | Top-*p* sampling (*p*=0.95) | 40.16 %       | 43.06 %            |
>
> The match rate comparison between greedy and other methods (Top-*k*, Top-*p*) showed insignificant differences in the summarization task. Despite minor variations in the QA task, non-greedy methods maintained a high match rate. These results suggest that applying PPD to other decoding algorithms could potentially enhance speed. Futhermore, we will clarify this matter explicitly in the revised manuscript. We will also provide the additional experimental results w.r.t. varying decoding strategies.
>
> **Response to "It lacks actual implementation"**
> We carried out a preliminary experiment to determine if our method could improve decoding speed. The results demonstrates that PPD indeed operates at a faster speed compared to greedy decoding. However, we acknowledge that the PPD comes with communication overhead. We are actively working on solutions to mitigate these issues in our future research. We report the details in the **General response** section.

---

### Review · Reviewer_gPqv · 2023-11-18

**Summary Of Contributions:**

The paper proposes a technique to reduce LLM token latency in greedy decoding by setting up a pipelined token computation which leverage the predictions of intermediate layers. This results in a trade off between the average per-token latency and the computing resources required to parallel-process the additional tokens: the more "next-next-token predictions" one wants to generate simultaneously, the more computationally expensive the process but the more accurate as well, which leads to decreased latency (theoretically).

The authors set up a theoretical framework to estimate latency vs. resource utilization as a function of the probability of correct predictions. Then extract this probability empirically, using Vicuna-13B on 3 tasks (QA on SQuAD1.1, translation on WMT EN-FR, summarization on CCN/DM). The estimated probability is plugged into their framework and the outcome discussed.

**Audience:**

Yes

**Broader Impact Concerns:**

Not mentioned in the manuscript. No concern on my end.

**Claims And Evidence:**

Yes

**Requested Changes:**

I don't have specific changes to request, I am just not entirely convinced of the benefits and applicability of the proposed method in real-world scenarios.

Minor clarifications and typos:
- section 4.2 and Figure 2 mention "total per-token latency"; if I am not mistaken, in this context it should read "total latency" (not normalized by sequence length)
- section 4.3: geration --> generation

**Strengths And Weaknesses:**

Strengths:
- the topic of LLM inference acceleration is of clear interest to the scientific and the industrial community alike, and highly relevant to this venue
- the paper is well structured and easy to follow
- the proposed idea is novel and the frameworks that links latency reduction to computational budget appears to be correct
- unlike other competing methodologies to reduce latency, this method provides the exact same output as the original LLM
- the probabilities of correct predictions to plug into their framework are extracted on a relevant LLM across a reasonable variety of tasks

Weaknesses:
- the speed ups appear modest in comparison to the additional resource requirements, even in the most optimistic scenario herein described (see next point). I am not sure what application would benefit from, for example, a 30% latency reduction at the cost of 3x compute resources to dedicate to the task. Could the authors provide some examples in support of the relevance of this technique?
- this is an "empirically-supported theoretical work". I appreciated the authors were very clear in stating that the latency vs compute resources curves are theoretical, and they also briefly mentioned as limitations of this study some potential overheads. Nevertheless, in my view, this remains one major limitation of this paper: the results may be entirely invalidated if the overheads associated to this method turn out to compensate (in part or entirely) the latency gains in practical applications. In this sense, the results presented herein are an upper bound for inference speed up (at each given additional computational budget). Some experimental data in this direction would make it for a much stronger paper.

---

> ### Author Response · Authors · 2024-01-17
> **Author's Response to Reviewer gPqv**
>
> We do appreciate your constructive feedback, and we hope our response fully addresses your concern.
>
> **Response to "The benefits and applicability of the proposed method in real-world scenarios"**
>
> The method can be effectively applied in scenarios where there is a need for numerous quick sequential inferences with a limited number of simultaneous calls. For example, consider a real-time financial service, such as stock price prediction program for an expert. In this situation, it is crucial for the Language Model to provide updates about the changing real-time financial status since even a marginal gain in latency could lead to a significant profit in this field. To increase the inference speed, it may be beneficial to utilize all available GPU resources.
>
> **Response to "Major limitation is that the results could be invalidated if the associated overheads offset the latency gains in practical applications"**
>
> We acknowledge that potential overheads could diminish the benefits of our method. Therefore, we have implemented the algorithm and shown that PPD has demonstrated its potential to enhance speed. You can see the experimental results in **General response**. Furthermore, we plan to revise the formula to incorporate real-world communication costs.
>
> **Typos**
>
> In response your feedback, we have corrected typos in the revised manuscript.

---

### Review · Reviewer_3gY2 · 2024-01-03

**Summary Of Contributions:**

Apologies for the late review.

The paper introduces a technique that, in theory, can reduce the latency of decoding/sampling in LLMs at the expense of increased compute. Extra compute appears in the form of $k$ subprocesses spawned to anticipate the future layer calculations of the next predicted token; done by passing the intermediate representation of some layer $\bar{d}$ through a soft-max classifier. The trade-off between compute increase and latency reduction is characterized analytically under the assumption that the matches (i.e., that the next predicted token is within the top-k predictions of the soft-max classifier) are iid.

**Audience:**

Yes

**Broader Impact Concerns:**

No concerns.

**Claims And Evidence:**

No

**Requested Changes:**

- I strongly recommend the authors implement the method or, at least, think about and discuss what overheads are expected in an implementation in more detail.
- Figure 1 is very nice and I found that Algorithm 1 was not very helpful. I would recommend presenting Algorithm 1 under the assumption that $\bar{d}=0.5d$ and $d$ is even. On this note, are the predicates for the while loop on line 6 correct, or should it be an *and* instead of *or*?
- In section 4.2 the line breaks when defining $\ell = N\cdot\sum_i X_i/N \approx N E\[X_1\]$ and the \cdot can be perceived as a period. This made me think, initially, they were 2 separate statements: $\ell = N$ and $\sum_i X_i/N \approx N E\[X_1\]$, which is confusing.

**Strengths And Weaknesses:**

I'm not an LLM expert and have not read any of the papers in the related work section.

# Strengths
Applying ideas of early-exiting to LLMs is interesting and can lead to significant speedups in decoding.

The method is presented very clearly (especially Figure 1 and the overall latency analysis of section 4).

Derivations are easy to follow and look correct.

# Weaknesses
The main issue with this paper is that the method was not implemented (as pointed out by the authors), leaving very little content in the paper.

Section 4.3 estimates the "match rate", but without a proper implementation using hardware acceleration (e.g., GPUs and TPUs) all results must be taken as best-case and theoretical.

It is not clear if the overhead in spawning subprocesses on GPUs will be outweighed by the theoretical gains presented in Table 2 as well as the appendix. For example, won't this method require moving extra memory between the on-device RAM and the shared memory space of the thread blocks? For transformer architectures this is a speed bottleneck.

---

> ### Author Response · Authors · 2024-01-17
> **Author's Response to Reviewer 3gY2**
>
> We do appreciate your valuable comments and suggestions, and we hope our response fully addresses your concern.
>
> **Response to "The method was not implemented"**
>
> Based on your recommendation, we have included the results of the PPD implementation in the **General response** section. We have found that PPD could truly lead to a speed increase. However, there is an overhead associated with multiprocessing.
>
> **Response to "... without a proper implementation using hardware acceleration (e.g., GPUs and TPUs) all results must be taken as best-case and theoretical."**
>
> We agree that the technique of spawning subprocesses on GPUs needs the transfer of additional memory between the on-device RAM and the shared memory space of the thread blocks. This could potentially result in a performance bottleneck, particularly for memory-intensive transformer architectures. Therefore, we will update the formula in our paper to accurately represent the communication cost incurred during memory transfer between processes.
>
> **About the Requested Changes**
>
> As per your great suggestion, we have seamlessly connected the lines in Section 4.2 you mentioned in the revised manuscript and will update Algorithm 1 description considering $\overline{d}=0.5d$.

---

### Author Response · Authors · 2024-01-17
**General response**

(R1 = R-3gY2, R2 = R-gPqv, R3 = WrQN)

We sincerely thank the reviewers for their thoughtful and constructive feedback.
We appreciate that the reviewers' acknowledgment of our interesting application of early-exiting ideas to Large Language Models (R1, R3), the well-structured format of our paper (R2), and the novelty of our proposed method (R2).

As for the concerns/questions raised, we believe that we successfully addressed all of them sufficiently and replied in line with each review.

**Common critique: It lacks the implementation of the proposed method**
We acknowledge that the theoretical work presented in the paper does not fully address the potential overheads that may invalidate the latency gains in practical applications. We have implemented the proposed method and conducted some preliminary experiments to provide empirical evidence supporting the efficacy of the method.


To measure actual latency, we have utilized the LLaMA-2 13B model and conducted tests on several examples from Summarization (CNN/DM) and Question Answering (SQUAD 1.1) tasks. We compare the speed of greedy decoding w/o PPD and that of greedy decoding w/ PPD. Additionally, we calculate the average latency and throughput when the LM generates 128 tokens.


### **Summarization**

| Method    | Match Rate| Latency   | Throughput  |
| --------  | --------  | --------  | --------    |
| Greedy w/o PPD (*k=0*, total 1 GPU)|  -        | 18.171 s  | 7.044 tok/s |
| Greedy w/ PPD (*k=1*, total 2 GPUs) | 25.72 %   | 17.019 s  | 7.521 tok/s |
| Greedy w/ PPD (*k=3*, total 4 GPUs) | 41.99 %   | 16.685 s  | 7.671 tok/s |


### **Question Answering**

| Method    | Match Rate| Latency   | Throughput  |
| --------  | --------  | --------  | --------    |
| Greedy w/o PPD (*k=0*, total 1 GPU)    |  -        | 14.994 s  | 8.537 tok/s |
| Greedy w/ PPD (*k=1*, total 2 GPUs) | 30.97 %   | 13.711 s  | 9.336 tok/s |
| Greedy w/ PPD (*k=3*, total 4 GPUs) | 47.24 %   | 13.712 s  | 9.335 tok/s |

From the results of the experiment, we observe that PPD potentially operates at a faster speed compared to the greedy decoding w/o PPD. However, it was found that there is no gain in latency when we compare the results of *k=1* to that of *k=3* in the Question Answering task due to the communication cost between processes.

Given these results, our future research will aim to address the limitations related to communication cost, with the goal of further enhancing the speed of PPD. Additionally, we intend to release the execution code for PPD upon acceptance of this paper. We have uploaded the code for an implementation of the proposed method in the supplementary material.

---

> ### Comment · Reviewer_gPqv · 2024-01-24
> **changes not included in the revised paper?**
>
> The latest uploaded manuscript does not seem to include the revisions made by the authors (including no mention of the PPD algorithm implementation on Llama_v2-13B and related results). Is this an issue with the upload or on my end?

---

> > ### Author Response · Authors · 2024-01-26
> > **Author's response to paper revision**
> >
> > We apologize for not properly checking the upload of the revision. We have re-uploaded it now for your review.

---

### Decision · Action_Editor_fpe2 · 2024-02-13

**Recommendation:** Accept with minor revision

**Comment:**

I am happy to accept this paper for publication in TMLR, but I agree with Reviewer 3gY2 that the paper must be clearer up-front about the limitations of the analysis, specifically that it fails to account for some overheads that, at least in the initial implementation, definitely matter in practice and reduce the speedup that can be achieved through PPD. Currently this is first explained in Section 5, but in fairness to the readers of this paper, it really should be disclosed in the abstract and introduction.

Note that the final recommendations from Reviewers 3gY2 and WrQN were received before the authors submitted the correct revision of the paper that includes the changes they made based on the initial reviewer feedback, so they are a bit too negative.

**Audience:**

All three reviewers (and I) agree that the paper will be of interest to some readers of TMLR.

**Claims And Evidence:**

The primary claims of the paper are (1) that the predictive pipelined decoding algorithm it proposes can be used to reduce the latency of LLM decoding at the cost of requiring increased computational resources and (2) the predictive pipelined decoding algorithm produces exactly the same output as standard decoding. The first claim is supported by a theoretical analysis of the algorithm in terms of the probability that the early exit prediction matches that of the full model; by experiments that estimate this match rate on three datasets for greedy, top-k, and top-p decoding; and by latency measurements on an initial implementation of the algorithm. The second claim is supported by the description of the PPD algorithm. While Reviewer WrQN contends that the PPD algorithm is "fundamentally similar to speculative decoding and should be compared with it," Reviewer gPqv argues that PPD is "potentially orthogonal to related techniques for speculative decoding." Having read the paper myself, I agree with the authors and Reviewer gPqv that the method differs significantly from speculative decoding. I do agree with Reviewers gPqv and 3gY2 that the current implementation results show the PPD algorithm has limited practicality because the latency reductions come at a substantial cost, but I also accept the authors' arguments that the implementation can likely be improved and that there may be specific applications where users may already be willing to accept the added computational costs for the sake of reduced latency.

---

> ### Author Response · Authors · 2024-03-13
> **Reply to Action Editors**
>
> Thanks for your constructive comments. We have included the information from section 5 in the abstract and introduction. In addition, we have uploaded our paper with code files and a video link.